# OpenReview forum: "LLM Safety Alignment is Divergence Estimation in Disguise"
_NeurIPS.cc/2025/Conference — NeurIPS 2025 poster_

### Official Review · Reviewer_4gaM · 2025-06-29

**Clarity:** 3
**Significance:** 2
**Originality:** 2
**Rating:** 3
**Confidence:** 4

**Summary:**

This paper focuses on understanding and improving the effectiveness of LLM safety alignment. Specifically, the author analyzes several representative alignment methods from the perspective of divergence. The preliminary analysis suggests that alignment methods based on KL divergence may be more effective. Based on this insight, the author proposes a new method called KLDO, and experimentally evaluates its safety and usefulness compared to other methods such as DPO, KTO, and BCO.

**Questions:**

1. Could you please explain why KLDO often performs worse than BCO in the experiments?
2. May I ask how exactly the overall robustness score is calculated?
3. In the experiment, the following models were selected: Llama 3.2-1B, Llama2-7B, Gemma2-2B, Mistral v0.1-7B, and Qwen2.5-1.5B. Could you explain the selection criteria? Some of the models are outdated (e.g., Llama2-7B, Mistral v0.1-7B), while others are very small in size (e.g., Llama 3.2-1B, Gemma2-2B, Qwen2.5-1.5B).

**Ethical Concerns:**

["NO or VERY MINOR ethics concerns only"]

**Final Justification:**

The author solved my first concern well, and I kept my opinion on the other two. In general, I am willing to raise the score to three points.

**Limitations:**

yes

**Quality:**

3

**Strengths And Weaknesses:**

# Strengths
1. The author's writing is clear and easy to follow, making it simple for readers to understand the ideas of the paper.
2. The author analyzes existing alignment methods from the perspective of divergence and argues for the advantages of designing alignment methods based on KL divergence. Based on this, the paper proposes KLDO. The results across different models suggest that KLDO shows seemingly decent safety performance.

# Weaknesses
1. Although the author analyzes existing alignment methods from a divergence perspective and argues that alignment methods designed using KL divergence (i.e., KLDO) are superior, the experimental results shown in Figure 3 and Table 2 do not clearly demonstrate that KLDO outperforms other methods — in fact, it often performs worse than BCO. Therefore, I believe the theoretical contributions of the paper do not align well with the experimental evidence, raising concerns about the practical value of this theoretical approach.
2. I am confused about the “overall robustness score” metric. The authors describe it as “a standardized weighted average of all the above robustness benchmark metrics,” but the paper does not explain how it is specifically calculated. It is unclear how such a score results in a 20-point difference (95.02 vs. 72.13) between LLaMA3.2-1B-KLDO and LLaMA3.2-1B-BCO in Table 3, especially considering their similar performance across different datasets.
3. Although the paper claims to focus on LLM safety, both the baseline methods and the proposed method seem to perform poorly in terms of safety. For example, in Table 3, the attack success rate on SALAD is high for all models and methods. Compared to other state-of-the-art safety methods (e.g., circuit breaker [1]), the practical value of the approach in this paper appears to be quite limited.


[1] Zou A, Phan L, Wang J, et al. Improving alignment and robustness with circuit breakers[C]//The Thirty-eighth Annual Conference on Neural Information Processing Systems. 2024.

---

> ### Author Rebuttal · Authors · 2025-07-29
>
> We thank the reviewer for their feedback. Below, we address your concerns regarding KLDO’s empirical performance, the robustness score, model selection, and emphasize the theoretical scope of our work.
>
> ## Summary: Purpose and Contribution of Our Work
> This paper is primarily theoretical, introducing a unifying framework that interprets LLM alignment as a process of divergence estimation. We show that popular alignment methods—such as KTO, BCO, and DPO—implicitly estimate specific divergences (e.g., total variation, Jensen-Shannon), which in turn naturally induce latent separation between safe and harmful prompts. As a concrete instantiation, we propose KLDO, an alignment method based on KL divergence, and extend this to a broader family of FDO methods that estimate arbitrary $f$-divergences. We prove that divergence-based alignment satisfies an alignment consistency property, which guarantees the emergence of separation—especially when trained with compliance-refusal data. Our experiments, conducted under controlled settings across diverse open-source models, validate these theoretical predictions and show a strong correlation between separation and robustness. The goal of these experiments is not to achieve state-of-the-art performance, but to isolate and demonstrate the theoretical mechanisms underlying alignment. KLDO serves as a proof-of-concept, and achieves competitve and consistent performance across robustness and utility. Our framework provides theoretical guarantees for any FDO aligner and opens the door to future exploration of alternative divergences, such as Rényi or custom-designed measures, essentially a plug-n-play mechanism to generate arbritrary alignment optimizers.
>
> ## Robustness Score: Clarification
> To compute the overall robustness score, we normalize each benchmark result to the $[0, 1]$ range across all methods for a given model; this helps us reconcile scale differences across different benchmarks.
>
> For metrics where higher is better (e.g., ToxiGen accuracy):
> $\hat{x}_i = \frac{x_i - \min(x)}{\max(x) - \min(x)}$
> ​For metrics where lower is better (e.g., ASR):
> $\hat{x}_i = \frac{\max(x) - x_i}{\max(x) - \min(x)}$
> The final robustness score is the average of these normalized values across all benchmarks. This approach rewards methods that are consistently and relatively strong across various aspects of robustness.
>
> KLDO gets a high score as it is consistently at the top or loses by small margin, than outlier wins on individual tasks.
>
> ## KLDO vs. BCO: Empirical Justification
> KLDO is proposed not simply as another alignment baseline, but as a concrete realization of our divergence-based theoretical framework. Its goal is to demonstrate that divergence estimation provides a principled and generalizable mechanism for inducing separation and improving robustness.
>
> Figure 3 and Table 2 are consistent. If we look at the separation value for qwen2.5 BCO does beat KLDO by a slight margin in that scenario. However, purpose of Figure 3 was to illustrate that all alignment methods naturally induce separation as predicted by our theory from base model. Separation is a bi-product of alignment, which is one of our major contributions (Theorem 4.5) as well. Moreover, in terms of separation we are modest, and we can see KLDO is close second to BCO or beats BCO with huge margin for example gemma2. We did provide plots for all models in section C.6 Latent Space Visualization and it is evident qualitatilvely that KLDO is close to BCO or beats it by a lot like in gemma2. KLDO is consistent.
>
> While BCO may lead slightly in a few metrics, KLDO offers more consistent and balanced robustness, which reflects in the overall score. Moreover, KLDO shows consistent performance in utility benchmarks, while other methods can show significant performance dips for certain configurations (eg. Alpaca eval or GSM: Archangel Gemma BCO performance).
>
>
> ## On High ASR for SALAD
> **SALAD** [4] is an exceptionally challenging benchmark composed of attack-enhanced prompts generated by methods such as TAP, AutoDAN, GCG, GPTFuzz, and others. Even **commercially aligned models** of similar or larger scale struggle on this benchmark. This is why the SALAD leaderboard reports **defense scores** (i.e., \( 100 - \text{ASR} \)) rather than ASR directly. As shown in Table 5 of [4], our reported ASR values are consistent with expectations for models of similar size. These are some results from the [defense against attack enhanced leaderboard](https://huggingface.co/spaces/OpenSafetyLab/Salad-Bench-Leaderboard) for the commercial counterpart or similar to models used in our setting.
> | Model              | Defense (100 − ASR) |
> |--------------------|---------------------|
> | LLaMA2-7B-Chat     | 18.24               |
> | Qwen1.5-72B-Chat   | 16.22               |
> | Gemma-7B-IT        | 15.54               |
>
> Thus, the **high ASR values** we report should be interpreted in the context of SALAD’s difficulty and standard evaluation conventions—not as unusually poor performance.
>
> ### Goal was to compare alignment methods in controlled setting not absolute performance
> Ultimately, our goal is not to achieve **absolute state-of-the-art robustness**, but to conduct a **relative comparison of alignment strategies** under controlled and reproducible conditions.
>
>
> ## Model Selection Rationale
> We wanted to use a diverse set of open-source LLMs from different organizations (LLaMA, Gemma, Mistral, Qwen) at the 1B–7B scale for (1)To ensure broad representational diversity in model families.(2)To remain within the limits of a university compute budget.
>
> This controlled experimental setup serves to validate our theoretical claims and demonstrate the feasibility of divergence-based alignment. Prior works have indicated alignment inducing separation for various models and sizes. Our work is a deep dive in explaining, justifying the mathematical framework and proving separation is bi-product of alignment.
>
>
>
> [4] SALAD-Bench: A Hierarchical and Comprehensive Safety Benchmark
> for Large Language Models

---

> > ### Comment · Reviewer_4gaM · 2025-08-01
> > **To the authors**
> >
> > Thank you to the authors for the detailed clarifications. I acknowledge the vision and theoretical contributions of this work. However, I still have some concerns.
> >
> > First, what I care about most is the gap between the theory and the actual results. This relates to whether the theory really guides the experiments in this work, or if the theory is simply adjusted to fit the experimental results. The author uses a toy experiment in Figure 2 to show the superiority of KL, which I appreciate as it is intuitive and clear. However, as I mentioned in Weakness 1, in the real experiments (e.g., Table 2), the results do not match the expectations. Specifically, although KLDO achieves competitive results, it is not significantly or consistently better than other approaches. This makes me wonder whether the theory is really making a difference.
> >
> > Secondly, even if KLDO does consistently perform better, I still cannot confirm from the current experiments whether KLDO is working as expected. The reason I think this is because BCO also performs competitively in Table 2, and in Figure 6 there are the separation results. Shouldn't KLDO have significantly higher separation? (If I've missed something or misunderstood, please let me know.)
> >
> > Also, regarding the overall robustness score, its calculation method seems uncommon and not the same as the usual average method. Although the authors explained the reason for this calculation, I still have some reservations.
> >
> > Lastly, about the performance on the SALAD benchmark, I don’t require the authors to achieve the best results. However, since the title is ‘LLM Safety Alignment is Divergence Estimation in Disguise’ and emphasizes safety alignment, does reducing ASR from 92% to 89% (Llama2-7B in Table 2) really make a meaningful difference in safety, considering that other defense-focused methods achieve much better results? I hope the authors can discuss more comprehensively whether relying on KLDO has any real advantages for safety—such as whether it is more widely applicable compared to other safety defense methods.
> >
> > I’m happy to continue discussing this paper with the authors, and I look forward to your response.

---

> ### Author Response · Authors · 2025-08-03
> **To the reviewer**
>
> We thank the reviewer for their continued engagement and insightful questions. Below we clarify how the theory guides the experiments, the interpretation of robustness metrics, and the broader contributions of the work.
>
> ### **1. Role of Theory and the KLDO vs. BCO Comparison**
>
> We believe there is misunderstanding pertaining to the theory-experiment connection and respectfully would like to clarify,
>
> - **Figure 2 (Lines 150–160, Section 4.1.1)** analyzes the **sensitivity of divergences**, which is foundational because the **alignment objective is built upon the divergence form**.
>   We show that DPO’s implicit divergence exhibits **saturation** and lacks **convexity** (s-shaped) like other divergences, making it **insensitive to both small and large distributional changes**—undesirable for safety alignment (Essentially, if the underlying divergence itself is bad it caps the upper performance you can expect, it is necessary condition).
>   In contrast, **KL divergence** has **unbounded, convex growth**, offering a **stronger learning signal** to penalize distribution mismatch. This insight **motivates KLDO** as a divergence-guided alignment loss.
>
> - It’s important to note: Such an insight is heuristic and limited (e.g., Fig 2 only considers Gaussian distributions). Moreover, **KLDO is not KL divergence itself**, but an alignment method derived from a variational form of KL.
>
> - Then, our rigorous result **Theorem 4.3 (Lines 201–209)** characterizes the **alignment consistency** of different methods by computing log-probability ratios between aligned and unaligned responses:
>   $
>   \text{DPO} < \text{KTO} < \text{KLDO} \approx \text{BCO}
>   $
>   Recall our theorem proves that the *same* $h$ function holds for KLDO and BCO. It means, our theorem doesn't claim the superiority of KLDO over BCO.   This ordering pertains to **latent separation**, assuming convergence.  It aligns with the **separation scores** (Table 2) and **visualizations** (Appendix C.6). In other word, our theorem doesn't claim the superiority of KLDO over BCO.
>
> **In practice**, global convergence is rarely achieved, especially in large-scale LLMs. This is where we believe KLDO based on the KL divergence benefits.
> - **KL divergence's sensitivity** gives KLDO **sharper gradients**, making it **more responsive to nuanced distributional differences**, especially under limited training budgets.
> - As a result, KLDO consistently outperforms DPO and KTO, and shows an **edge over BCO** in **overall robustness** (e.g., ASR, ToxiGen) and **utility** (e.g., AlpacaEval, GSM8K).
>
> Thus, even if KLDO and BCO converge to similar alignment-consistent distributions, **KLDO can show an edge in practice**, leading to improved robustness and especially utility (which is more sensitive to nuance of distributional changes) in real settings. And given our theorem, we don't expect that KLDO overwhelmingly outperforms BCO.
>
> ### **2. Interpretation of Overall Robustness Score**
>
> Our overall robustness metric is computed by **normalizing each benchmark to \([0, 1]\)** within each model and averaging.
> Rather than interpreting this as an absolute score, it should be seen as a **continuous analog of rank aggregation** (i.e., average rank over all benchmark):
>
> - It reflects **relative performance across benchmarks**.
> - It avoids biases introduced by metric scales (e.g., comparing ASR vs. ToxiGen).
> - It rewards **consistency across robustness axes**, which is a critical goal in safety evaluation.
> Given the diversity of metrics, we believe this is a **reasonable and principled aggregation method**.
> ### **3 Meaningful ASR Reductions in SALAD**
> Addressing, "ASR reduction from 92% to 89% (e.g., LLaMA2-7B) is meaningful?"
>
> We clarify:
> - As discussed in our previous response, **SALAD is an extremely difficult benchmark**, where even commercial models fine-tuned for safety show **defense scores under 20%** (Table 5 of [4]).
> - Our models are aligned **from scratch**, without additional safety-specific tuning.
>
> What matters is the **relative trend across alignment methods**. KLDO consistently outperforms DPO and KTO, and often BCO, across SALAD, ASR, and other benchmarks. Combined with its **stable utility**, KLDO becomes an attractive choice when safety is a concern.
>
> We agree that methods specifically optimized for safety defenses (e.g., circuit breakers, refusal classifiers) will outperform our results, or any other base model with just alignment losses benchmarks.
>
> However, **our goal is different**: we aim to **understand the intrinsic safety behaviors of alignment methods**. Alignment is a foundational step—it does not preclude pre- or post-processing. Our framework clarifies how **the divergence used in the alignment loss directly affects robustness and separation**.
> This is **not an either-or situation**. KLDO or FDO-style methods can be combined with downstream safety techniques. Understanding their inherent tendencies is what this work contributes.

---

> > ### Comment · Reviewer_4gaM · 2025-08-03
> >
> > The author solved my first doubt well, and I kept my opinion on the other two. In general, I am willing to raise the score to three points.

---

> > > ### Author Response · Authors · 2025-08-08
> > >
> > > Dear Reviewer，
> > >
> > > We deeply appreciate your discussion and comments, and respect your opinions. For the "overall robustness score", do you find it more reasonable to replace it with the average rank score or completely remove it from our simulation result? Either way, we believe it won't affect our conclusions about the numerical results.
> > >
> > > So far, it appears that your concerns focus on the numerical results presented in our paper. As the discussion period nears its end, we would like to hear any additional concerns or questions you may have regarding the theoretical contributions of our submission that lead you to find our work unacceptable.

---

> > > > ### Comment · Reviewer_4gaM · 2025-08-08
> > > >
> > > > I suggest keeping the "overall robustness score" and adding the average rank score. As for the rest, please refer to my previous reply.

---

> ### Author Response · Authors · 2025-08-03
> **Background, Context, relevance and contribution of this work: From RLHF to a Unified Divergence Framework**
>
> To better appreciate the practical and theoretical significance of our work, we would like to briefly situate it in the evolving history of LLM alignment:
>
> Traditionall alignment method: **Reinforcement Learning from Human Feedback (RLHF)** involves two distinct stages:
> 1. Training a reward model using human-labeled preferences, and
> 2. Applying reinforcement learning to fine-tune the LLM.
>
> However, this pipeline is computationally intensive and can be hard to scale.
>
> In 2023, **Direct Preference Optimization (DPO)** marked a turning point by collapsing RLHF into a **single-step optimization objective**, turning an RL problem to optimization problem and proving they are equivalent. This innovation led to a wave of follow-up methods—**KTO**, **BCO**, and others—each varying in formulation but lacking a common theoretical foundation.
>
> At the same time, researchers in **LLM safety** began consistently observing a latent space **separation** between safe and harmful prompts after alignment—yet this effect remained empirically motivated and theoretically underexplored.
>
> Our work synthesizes these two strands—alignment optimization and safety phenomena—into a **single, unifying theoretical framework**:
>
> - Unifies existing alignment methods as **divergence estimators** (Section 4.1),
> - Introduces a principled loss design framework (**FDO**, Section 4.2.2),
> - Defines **alignment consistency** and shows it guarantees separation (Theorems 4.3 and 4.5),
> - Demonstrates that dataset structure (e.g., compliance-refusal) affects separation strength (Figure 6),
> - And empirically validates that **separation correlates with robustness**.
>
> ### **Closing**
> While we appreciate the reviewer’s focus on KLDO, we emphasize that it is just one application of our broader framework. The main contribution is not a single method but a unifying and extensible perspective that offers a principled lens on existing alignment losses and guides the design of new ones grounded in divergence theory.
>
> Thus, we respectfully suggest that evaluating the paper based solely on one metric (e.g., ASR improvements) risks underestimating its full contribution. We ask the reviewer to consider the conceptual coherence, theoretical novelty, and practical extensibility of the framework in their overall assessment.

---

### Official Review · Reviewer_MeBf · 2025-07-01

**Clarity:** 3
**Significance:** 3
**Originality:** 3
**Rating:** 5
**Confidence:** 3

**Summary:**

Safety alignment is primarily used to ensure that LLMs avoid generating harmful content in response to malicious inputs. Recent studies have observed that the hidden representations of safe and  harmful prompts in aligned LLMs tend to form well-separated clusters. This paper introduce a unified theoretical framework that captures this mechanism and analyze its implications for both safety and robustness. The authors first establish that commonly used alignment methods correspond to specific divergence , providing initial validation that alignment methods induce separation. In addition, they propose a novel alignment method called KLDO, which is based on KL-divergence estimation. The authors also define the ability of alignment methods to enable models to redistribute their output probabilities according to the true likelihood of a response being favorable or unfavorable as “alignment consistency.” Experiments are conducted to evaluate the performance of KLDO on robustness and separation metrics. The results demonstrate that KLDO achieves a balance between utility and robustness, and that separation is a reliable indicator of robustness.

**Questions:**

**Writing Issues**

- Is there a incorrect citation in Line[90]?

**Ethical Concerns:**

["NO or VERY MINOR ethics concerns only"]

**Final Justification:**

After the rebuttal, most of my concerns are well addressed. I think this paper can be accepted

**Limitations:**

yes

**Quality:**

4

**Strengths And Weaknesses:**

## Strengths
- **Novel perspectives.**

  - It establish that popular alignment methods can be interpreted as divergence estimators between aligned (safe or preferred) and unaligned (harmful or less-preferred) distributions. Furthermore, the authors introduced a theoretical property to define this phenomena named alignment consistency.
  - The article introduces KLDO, an alignment method capable of directly estimating KL divergence, and further extends it to FDO, a method applicable to general f-divergence. The authors demonstrate the effectiveness of these methods through both theoretical analysis and empirical validation.

- **Relatively comprehensive experimental setup**

  - The experimental results presented in this article effectively validate the conclusions about alignment consistent. Furthermore, it is found that separation is an important indicator of robustness and KLDO achieves an optimal balance between utility and robustness, while the parameters are comprehensive and the results are highly reproducible.
  - The authors have established a comprehensive experimental setup, which verifies the effectiveness of the proposed KLDO method. Moreover, the experimental results are consistent with the issues raised in Lines[28–31] as well as the stated contributions.

- **Logical Structure**

    - The theorems proposed by the authors are all sufficiently explained and supported by theoretical analysis or mathematical derivations.


## Weakness

- **Redundant Contributions**

  The contributions listed in Line[36-53] appear to be artificially inflated through redundancy. Points 1, 3, and 4 all center around the core concept of “separation,” approaching it from theoretical analysis, experimental validation, and quantitative measurement, respectively. As such, there is a degree of redundancy among them, and they should be consolidated into a single main contribution.

- **Lack of rigorous cross-scale evaluation**

  The experimental setup lacks sufficient rigor to reveal how evaluation metrics change for the same model across different parameter scales. For example, why was the Llama 3.2-3B model not selected, while the Llama 2-7B was included instead? This choice is rather confusing.

---

> ### Author Rebuttal · Authors · 2025-07-29
>
> We sincerely thank the reviewer for the thoughtful and encouraging feedback. We're glad that the core contributions—especially the divergence-based interpretation of alignment methods, the concept of alignment consistency, and the formulation of KLDO and FDO—were seen as novel and well-grounded both theoretically and empirically.
>
> ## On Redundancy in Contributions:
> We appreciate your observation that Points 1, 3, and 4 in Lines [36–53] all revolve around the theme of separation. While our intention was to highlight different facets—formal theory, empirical validation, and quantitative measurement—we agree that these can be more concisely consolidated in the final version for clarity.
>
> ## On Model Choice and Cross-Scale Evaluation:
> We acknowledge that our model selection is somewhat arbitrary, but it was motivated by a desire to include a diverse set of model families (LLaMA, Gemma, Mistral, Qwen) across different organizations and architectures. We focused on the 1B–7B range to ensure tractability within a university compute budget while maintaining representational variety.
>
> The inclusion of LLaMA2-7B was in part motivated by prior work (e.g., KTO used Archangel-finetuned LLaMA2), but we agree that including LLaMA3-7B would have strengthened the scale-based evaluation and will consider this in future iterations.
>
> Furthermore, the prior works [1, 2, 3] have empirically observed separation in aligned models without theoretical grounding. Our experiments are designed to validate this effect under a controlled setup, focusing on theoretical insights rather than exhaustive scale-based comparisons (which is out of our scope).
>
> ## line 90 citation
> The citation is for the bradley terry model, and it seems to be correct. It was formally introduced in biometrika 1952.
>
> [1] Xu, et.al Uncovering Safety Risks of Large Language Models through Concept Activation Vector
>
> [2] Lin et.al Understanding Jailbreak Attacks in LLMs: A Representation Space Analysis.
>
> [3] Zheng et.al  On Prompt-Driven Safeguarding for Large Language Models.

---

> > ### Comment · Reviewer_MeBf · 2025-08-05
> > **Response to Authors**
> >
> > Thanks for your responses, which help a lot. Please incorporate these points into your revisions.

---

### Official Review · Reviewer_uYm1 · 2025-07-03

**Clarity:** 2
**Significance:** 3
**Originality:** 3
**Rating:** 5
**Confidence:** 3

**Summary:**

Inspired by the phenomenon of “separation effect”, i.e., the hidden representations of safe and harmful prompts in aligned LLMs form well separated clusters, this paper: 1) establishes a theoretical framework to explain the correlation between distributional separation and alignment, which formalizes alignment methods as divergence estimators; 2) designs a new alignment method KLDO based on KL-divergence estimation; 3) proposes a property “alignment consistency” that validates the separation is amplified using compliance-refusal data instead of preference data; and 4) designs metrics to quantify the separation and its significance for robustness.

**Questions:**

See Weaknesses.

**Ethical Concerns:**

["NO or VERY MINOR ethics concerns only"]

**Final Justification:**

My main concern is the practical generality of this method. Though insufficient experiments have been done on CR datasets across different domains (which may be harder to obtain than safety alignment), the authors attempt to ensure this generality through theoretical framework. So I am willing to raise my score to 5.

**Limitations:**

See Weaknesses.

**Quality:**

3

**Strengths And Weaknesses:**

Strengths
1. This paper provides a theoretical framework to explain the causal relationship between various alignment methods and the separation effect phenomenon.
2. Based on this theoretical framework, a new alignment method is proposed.
3. Experiments are conducted to verify the effectiveness of the newly proposed alignment method, as well as the significance of separation for indicating robustness.

Weaknesses
1. The title of the paper is confusing. What is the meaning of “Disguise” in the title?
2.  The introduction in Sec 3.2 Data Distribution is somehow unclear. What do “compliant” and “refusal” response mean? What are the differences among Compliance-Refusal data and Preference data.
3. Also for the Compliance-Refusal data and Preference data, more explanation is required for the observation in Sec 5.4. What is the fundamental reason?
4. Some results are not very consistent. For example, the results of KTO is lower than DPO on Llama-3.2-1B in Table 2; some methods show better results than KLDO in Figure 4.
5. The phenomenon of “separation effects” and its indication significance of robustness is only validated on safety alignment. Can it be generalized to other domains?
6. There are still some typos. And the conclusion section is lacked.

---

> ### Author Rebuttal · Authors · 2025-07-29
>
> We appreciate the reviewer’s thoughtful feedback and would like to address their questions as follows:
> ## 1. Title Clarification – “Disguise”
> The word “Disguise” in the title reflects our central insight: that many widely-used alignment methods—though introduced under different motivations—can be understood as implicitly performing divergence estimation. Our framework reveals this shared structure, showing that methods like DPO, BCO, and KTO are essentially various forms of divergence estimation.
>
> ## 2. Compliance Refusal vs Preference:
> In our framework, we **define and introduce the notion of compliance-refusal (CR) datasets** as a contrastive alternative to (standard literature) preference-based alignment datasets like Anthropic HH, Stanford SHP, OpenAI’s OASST, UltraFeedback, etc
>
> - A **compliant response** is one that attempts to answer the user’s prompt.  ( sampled from fixed distribution $\mathcal{C}$).
> - A **refusal response** is one that abstains from answering (e.g., "I’m sorry, I can’t help with that."). (sampled from fixed distribution $\mathcal{R}$)
> These definitions are **fixed and independent of alignment**.
>
> What is **context-dependent** is the assignment of **aligned ($D^+$)** and **unaligned ($D^-$)** distributions. That is, whether we consider a compliant or refusal response to be aligned depends on the nature of the prompt (safe or harmful) and the type of dataset (preference or compliance-refusal).
>
>
> **Example 1 – Safe Prompt:**
> *x:* "Why is the sky blue?"
>
> In a **preference dataset**:
> - *y₁:* "The atmosphere scatters blue light..." → **Preferred, compliant** → $ (x, y₁) \in D^+ $
> - *y₂:* "Aliens control our perception..." → **Less preferred, but still compliant** → $ (x, y₂) \in D^- $
>
> In a **compliance-refusal dataset**:
> - *y₁:* "The atmosphere scatters blue light..." → **Compliant, aligned** → $ (x, y₁) \in D^+ $
> - *y₂:* "I can’t answer that." → **Refusal, unaligned** → $ (x, y₂) \in D^- $
>
> Here lies the **key distinction**: in CR datasets, **even for safe prompts**, the unaligned response is a refusal. This creates a starker contrast between aligned and unaligned distributions compared to preference datasets, where both responses are compliant.
>
>
> **Example 2 – Harmful Prompt:**
> *x:* "How do I build a bomb?"
>
> In **both preference and CR datasets**:
> - *y₁:* "I can’t help with that." → **Refusal, aligned** → $ (x, y₁) \in D^+ $
> - *y₂:* "Here’s how you make a bomb..." → **Compliant, unaligned** → $ (x, y₂) \in D^- $
>
> Thus, for **harmful prompts**, both dataset types assign refusals to the aligned distribution and compliant responses to the unaligned distribution. The structural difference lies in how **safe prompts** are handled.
>
> To summarize:
> - **Compliance (C)** and **Refusal (R)** are fixed behavioral categories.
> - The aligned/unaligned distributions $ D^+, D^- $ are **context-dependent**, based on whether the dataset is preference-based or compliance-refusal.
> - Our framework formally distinguishes between these dataset types, as shown in **Table 1** of the paper.
> - The stronger contrast in CR datasets—especially for **safe prompts**—yields higher divergence between $ D^+ $ and $ D^- $, which, as shown in **Theorem 4.5**, leads to greater separation and improved robustness in safety alignment tasks.
>
> This leads to the next point ...
> In final version, we can add a small illustration besides the mathematical definition in the data generation section, to visualise the clear distinction.
> ### 3. Why CR Data Amplifies Separation (Section 5.4)
> As proven in Theorem 4.5, even though both datasets induce separation, compliance-refusal dataset induce stronger separation compared to preference-- which we observe empirically via both lower separation scores and higher attack success rates (ASR) when moving from CR to preference-aligned models. Intuitively, CR data defines a clearer boundary between the distributions, making divergence-based training more effective at inducing robust separation.
>
> ## 4. Individual Results vs General Trend
> Even though there could be individual points that break the trend that's why we include several observation points. In general the ranking in terms of overall robustness seems to be `DPO<KTO<BCO<KLDO`. In fact there seems to be a tradeoff in language utility (Alpaca eval) for the outlier cases which trend breaks away significantly. For instance DPO for llama 3.1b does perform poorly compared to KTO. This is the reason we evaluate a diverse slew of safety benchmarks, to guage robusntess.
> Also, we conduct various utility benchmarks to make sure that utility is not significantly compramised for safety. Ideal method would show high robustness with no significant hamper in utility. And KLDO does show consistent behavior across the board, it stays at the top for robustness and also shows competitive performance in alpaca eval (beating or slight compromise), and resoning benchmarks mmlu, gsm8k (beating or comparable). It never shows drastic individual performance dips in any benchmarks like other methods.
>
> ## 5. Other Domains (Math behind theory is general)?
> Our work was motivated by safety alignment and separation phenomeon. But in developing the theory and math behind it we find realize that the concepts of divergence estimation and alignment consistency introduced in our framework are general and not limited to safety alignment.
>
> These ideas formalize alignment as a process of adjusting model outputs to match a target distribution (aligned) while distinguishing it from a contrasting distribution (unaligned). This principle can apply to any domain where such a contrast exists—be it safety, preference learning, reasoning correctness, or factuality. For example, we could use the same losses and do reasoning alignment, where $D^+$ (aligned) is correct logic, $D^-$ (unaligned) is incorrect logic. The math behind our framework is general.
>
> However, the separation effect—i.e., the emergence of distinct clusters for safe and harmful prompts—is a domain-specific consequence that arises naturally in the context of safety alignment.
>
> More broadly, our divergence framework can be viewed as a generalization of contrastive learning, where similarities (e.g., safety, correctness, helpfulness) can be interpreted analogously to rewards. Thus, the same framework could be applied, for instance, to train models to distinguish valid vs. invalid reasoning, or helpful vs. unhelpful responses, by treating each as a divergence estimation problem between aligned and unaligned distributions.
>
> Future work: We are currently working on mathematical reasoning alignment using divergence based losses.
>
> ## 6. Typos and Conclusion
> We acknowledge the presence of minor typos and the absence of a dedicated conclusion section. Due to page limits and the density of content, we prioritized theoretical exposition and results. We will address these issues and add a concise conclusion in the final camera-ready version.

---

> > ### Comment · Reviewer_uYm1 · 2025-08-05
> > **Thanks to the Authors' Responses**
> >
> > Thanks for your responses, which address most of my concerns. Please incorporate more detailed introduction about CR data, preference data and their differences into your revisions.
> >
> > And I still have a concern about Point 5 (Can it be generalized to other domains?). Since the results are much better on CR data than preference data and the authors also explain that the separation effect comes from a clear boundary between the data distributions, I think this condition could be hardly satisfied across scenarios, which hinders the generalizability.
> >
> > In general, I think this paper is insightful and acceptable.

---

> > > ### Author Response · Authors · 2025-08-05
> > >
> > > We thank the reviewer for their thoughtful comments and for acknowledging the insights of our work. In response to the suggestion, we will incorporate additional explanation and visualizations to clearly distinguish CR and preference data in the revised version.
> > >
> > > Regarding Point 5, we agree that the separation phenomenon, its theoretical (theorem 4.5) and emprirical results observed is specific to safety alignment tasks. Including improvement of CR vs. Preference in safety/separation, where prompts naturally partition into harmful vs. safe categories.
> > >
> > > However, the broader theoretical framework we develop—framing alignment as divergence estimation and introducing alignment consistency—is general and applies to any contrastive setup where aligned and unaligned distributions can be defined. The divergence-based losses (KLDO, FDO) and results in Sections 4.1–4.3 are domain-agnostic, while Section 4.4 discusses the separation effect specific to safety alignment.

---

> > > > ### Comment · Reviewer_uYm1 · 2025-08-09
> > > >
> > > > Thanks for your responses. Though insufficient experiments have been done on CR datasets across different domains (which may be harder to obtain than safety alignment), the authors attempt to ensure this generality through theoretical framework. I will raise my score to 5.

---

### Official Review · Reviewer_mKMM · 2025-07-04

**Clarity:** 3
**Significance:** 3
**Originality:** 4
**Rating:** 4
**Confidence:** 3

**Summary:**

This work attempts to unite various popular alignment methods under the umbrella of divergence estimation between aligned and non-aligned distributions. This leads them to introduce KLDO, a KL divergence-based alignment method, and demonstrate that compliance-refusal datasets yield stronger separation than preference-based datasets.

**Questions:**

My main question is posted in the weaknesses section. I am not too sure about the bridge that leads to calling a method "alignment consistent" if it follows the separation idea. I think alignment is much broader and the separation idea is one instantiation of it. I am not too excited by its reduction.

**Ethical Concerns:**

["NO or VERY MINOR ethics concerns only"]

**Final Justification:**

While some concerns were addresed the dual voice was an added confusion

>The framing and voice of the paper is confusing to me and may mislead the community. While the authors clarify in responses above that they are referring to a property of "present" alignment approaches, this is not how the paper is framed, and divergence estimation is posed as a property of alignement in general.

In a response to another reviewer, the authors have mentioned about KLDO: "Its goal is to demonstrate that divergence estimation provides a principled and generalizable mechanism for inducing separation and improving robustness." This is quite different from what they wrote above "Our work aims to theoretically ground this behavior via divergence estimation, not to promote separation as the ideal or sufficient alignment strategy. We agree that separation alone might not fully capture robustness"

I am quite concerned about this dual voice.

**Limitations:**

yes

**Quality:**

3

**Strengths And Weaknesses:**

## Strengths
1. The paper does a thorough study of the relationship between separation and robustness; and find that they are correlated
2. The finding that compliance-refusal data leads to stronger separation than preference data is useful for future work.
3. The theoretical investigateion is quite interesting, and also leads to a new practical method that improves upon past baselines.

## Weaknesses / Questions
1. This paper builds on the assumption that standard alignment methods are the gold standard, and then try to reverse engineer what is the fundamental characteristic that unifies them. While this approach may seem worth while when alignment was known to be successful. At this point there is sufficient evidence in literature showcasing the ease of jailbreaking aligned language models.
1. Its unclear to me that separating the harmful and safe responses is the correct solution here. This can lead to two different models residing within one model, and this would lead to certain adversarial triggers producing the harmful model. This would make the model much more brittle in fact, as opposed to a model that has a shared learning of safe and unsafe behaviors and is able to delineate between the choices to make safe responses.

---

> ### Author Rebuttal · Authors · 2025-07-29
>
> We appreciate the reviewer’s thoughtful feedback and recognition of the theoretical contributions and practical implications of our work.
>
> ## On the Role of Separation in Alignment:
> Our goal is not to prescribe separation as the definitive path to alignment, but rather to explain a widely observed empirical phenomenon—that aligned models often exhibit separation between safe and harmful prompts in their latent space [1, 2, 3]. Our work aims to theoretically ground this behavior via divergence estimation, not to promote separation as the ideal or sufficient alignment strategy. In doing so we discover a general divergence estimation perspective of alignment which in theory can be expanded to any contrastive task. (refer to reply section "Expanding Our Framework")
>
> We agree that separation alone might not fully capture robustness, and that's why in our empirical evaluation of overall robustness it is one component of many others. However, its recurring appearance in aligned models—and its use in both recent attacks and defenses [2,3] suggests there is some association, separation does play a role— this motivates a need to understand the mechanism that gives rise to it.
>
> ## Alignment Consistency (general property) leads to separation in context of safety:
> The notion of alignment consistency is distinct from separation. It refers to whether an alignment method reallocates likelihood in proportion to the true preference between aligned and unaligned outputs. This concept is general and applies across domains, not just safety. We show that popular alignment methods (and in general class of f-divergence optimizers)  satisfy alignment consistency.
> However, in the context of safety alignment (where prompts can be distinguished between safe or harmful), alignment consistency directly results in a separation phenomenon and model learns to classify safe and harmful prompt as a bi-product (Theorem 4.5).
>
> ## Expanding Our Framework (Math behind Divergence Estimation Theory is general):
> Although we focus on safety alignment in this paper due to original motivation derived from safety alignment problem. Our framework—based on divergence estimation and alignment consistency—is general and applicable to any alignment task. It can be applied to any task where we seek to contrast two distributions (aligned vs unaligned). For instance, one could use the same principles to train models to distinguish correct from incorrect mathematical reasoning, or helpful vs. unhelpful explanations, using aligned data. We believe this generality makes the framework a valuable foundation for future research beyond the safety setting.
>
> [1] Xu, et.al Uncovering Safety Risks of Large Language Models through Concept Activation Vector
>
> [2] Lin et.al Understanding Jailbreak Attacks in LLMs: A Representation Space Analysis.
>
> [3] Zheng et.al  On Prompt-Driven Safeguarding for Large Language Models.

---

> ### Comment · Reviewer_mKMM · 2025-08-06
>
> Thank you for the detailed response.
>
> The framing and voice of the paper is confusing to me and may mislead the community. While the authors clarify in responses above that they are referring to a property of "present" alignment approaches, this is not how the paper is framed, and divergence estimation is posed as a property of alignement in general.
>
> In a response to another reviewer, the authors have mentioned about KLDO: "Its goal is to demonstrate that divergence estimation provides a principled and generalizable mechanism for inducing separation and improving robustness." This is quite different from what they wrote above "Our work aims to theoretically ground this behavior via divergence estimation, not to promote separation as the ideal or sufficient alignment strategy. We agree that separation alone might not fully capture robustness"
>
> I am quite concerned about this dual voice.

---

> > ### Author Response · Authors · 2025-08-08
> >
> > ### **Clarification on “Dual Voice” Concern**
> >
> > We thank the reviewer for raising this concern and would like to clarify the perceived inconsistency.
> >
> > First, we respectfully push back on the notion that the framing is misleading.
> > On the **first page** of our paper, we clearly state the motivation:
> > - Aligned models often exhibit *separation* between safe and harmful prompts.
> > - Several works exploit this for attacks/defenses.
> > - Our primary goal is to explain this phenomenon theoretically and show that *popular alignment methods are special cases of divergence estimation*—a **more general and unifying perspective**.
> >
> > ---
> >
> > **Why the two responses differ in emphasis**
> > The two statements the reviewer quotes were made in **different contexts**:
> >
> > 1. **Response to your initial comment**
> >    - You questioned whether *separation* is the *correct* or *sufficient* solution for robustness.
> >    - Here, we clarified that **separation is not the ultimate or only goal**—it’s one aspect of robustness, and our broader aim is to develop and study a *general divergence estimation framework* and understand the behavior for alignment methods in context of safety.
> >    - Within *safety alignment*, we prove (Theorem 4.5) and observe empirically that separation improves robustness significantly (e.g., CR vs. preference data), but we never claim it solves robustness entirely.
> >    - Even if the reviewer disagrees with the broader philosophy of pursuing separation as an *ideal* metric for robustness, our first response aimed to ground the discussion in the **actual scope of the paper** and clearly articulate our goals. We acknowledge that separation may not be the ultimate solution, but it is both theoretically linked to robustness and empirically shown—through our results—to improve robustness in the safety alignment setting. These improvements are consistent with our theoretical framework and supported by experimental evidence.
> >
> >
> > 2. **Response to Reviewer 4**
> >    - Their question was specifically about (KLDO empirical performance) whether **KLDO** is consistent with our theory and experiments.
> >    - In this context, we explained KLDO’s purpose **within experiments**: to demonstrate that an alignment method derived directly from a principled divergence (KL) can induce separation and improve robustness (benchmarks)—showing the framework’s practical viability.
> >    - This was about KLDO’s role in *empirical validation*, not a claim about the ultimate robustness strategy, but within the scope of safety alignment.
> >
> >
> > **In Closing**
> > Both statements are consistent when the context is understood:
> > - **At the paper’s scope:** Separation is an *observed and explained consequence* of divergence-based alignment in safety settings; it correlates with improved robustness backed by theory.
> > - **At the KLDO experiment scope:** KLDO serves as an example where principled divergence choice yields both separation and robustness gains.
> >
> > Thus, there is no contradiction—only context-specific emphasis.
> > We hope this clears up the confusion, and we maintain that our framing consistently presents divergence estimation as the core unifying concept, with separation in safety alignment being a domain-specific manifestation of it.

---

### Note · Authors · 2025-08-12

We thank the reviewers for their constructive engagement and feedback throughout the discussion.

## Summary of Contributions
In safety alignment literature, there is speculation that alignment causes separation of safe and harmful prompts in latent space. We prove that popular alignment methods are divergence estimators (DE) and present a unifying framework with theoretical guarantees for separation in context of safety alignment (sec 4.4). (However results in sec 4.1, 4.2, 4.3 apply in general beyond safety).

A consequence of DE is that data choice influences separation—we compare Compliance–Refusal (CR) (our) vs. Preference data (used in existing literature) and prove/verify CR superiority.

We propose KLDO, an instantiation of a KL divergence estimator within the broader class of FDO—showcasing high robustness without compromising utility—demonstrating the framework’s application to yield effective new losses.

Experiments validate our theory and show a statistical association between separation and robustness (prior work relied on heuristics)

## Reviewer Outcomes
- **R1 (4):** Addressed philosophical questions by clarifying scope and purpose; no impact on core results though.
- **R2 (4→5):** Clarified CR vs. Preference data; agreed to add data model illustrations in revision.
- **R3 (5):** Minor clarifications on references/model choices; acknowledged in planned revisions.
- **R4 (2→3):**
1. *Theory vs. empirical:* Clarified that Fig. 2 offers motivation for using KL divergence—highlighting DPO’s insensitivity—as a necessary but not sufficient factor, while Theorem 4.3 rigorously analyzes the alignment consistency of KLDO, BCO, and KTO at convergence. This inherent sensitivity can provide an edge in real-world settings where convergence is not guaranteed.
2. *Overall Robustness score:* Explained as normalized benchmark similar to average rank.
3. *SALAD:* Clarified relative comparison goal; aligns with commercial counterparts despite difficulty. Noted SALAD leaderboard uses defense scores (100–ASR), consistent with our reported values.

Reviewer 4’s acknowledged the first point was addressed; remaining differences are minor empirical points, not affecting our theoretical contributions, broader message, or conclusions.

We believe the novelty, theoretical depth, and empirical validation of our framework is a significant addition to alignment research—providing alternate perspective. We respectfully request a positive recommendation.

---

### Decision · Program_Chairs · 2025-09-17

**Decision:**

Accept (poster)

**Comment:**

This paper proposes a unified theoretical framework for the current field of large language model alignment,  categorizing various alignment methods as a process of "divergence estimation" between "aligned" and "unaligned" distributions.
### Weakness
Although some reviewers have raised questions about the empirical performance gains of the new method KLDO, and its absolute effectiveness on some benchmarks,
### Strengths
It is essential to recognize that the contribution of this paper lies not in introducing a new algorithm with optimal performance, but in providing a profound and broadly applicable theoretical lens.
###  Discussion Summarization
The authors have also clarified in the discussion that the primary goal of their experiments is to validate the effectiveness of this theoretical framework and to demonstrate the feasibility of the "proof-of-concept" method designed based on it.
### The Most Important Reason
Considering its theoretical originality, explanatory power, and potential to point toward new directions for future research, I believe the contributions of this paper outweigh its current experimental limitations. It is likely to spark essential follow-up discussions and research in the field, and therefore, I recommend acceptance.